# ESCRT-III-driven piecemeal micro-ER-phagy remodels the ER during recovery from ER stress

Marisa Loi[1,2], Andrea Raimondi[3], Diego Morone[1] & Maurizio Molinari ® [1,4]*

The endoplasmic reticulum (ER) produces about 40% of the nucleated cell's proteome. ER size and content in molecular chaperones increase upon physiologic and pathologic stresses on activation of unfolded protein responses (UPR). On stress resolution, the mammalian ER is remodeled to pre-stress, physiologic size and function on activation of the LC3-binding activity of the translocon component SEC62. This elicits recov-ER-phagy, i.e., the delivery of the excess ER generated during the phase of stress to endolysosomes (EL) for clearance. Here, ultrastructural and genetic analyses reveal that recov-ER-phagy entails the LC3 lipidation machinery and proceeds via piecemeal micro-ER-phagy, where RAB7/LAMP1-positive EL directly engulf excess ER in processes that rely on the Endosomal Sorting Complex Required for Transport (ESCRT)-III component CHMP4B and the accessory AAA$^+$ ATPase VPS4A. Thus, ESCRT-III-driven micro-ER-phagy emerges as a key catabolic pathway activated to remodel the mammalian ER on recovery from ER stress.

---

[1] Faculty of Biomedical Sciences, Institute for Research in Biomedicine, Università della Svizzera italiana (USI), Bellinzona, Switzerland. [2] Department of Biology, Swiss Federal Institute of Technology, 8093 Zurich, Switzerland. [3] Experimental Imaging Center, San Raffaele Scientific Institute, 20132 Milan, Italy. [4] School of Life Sciences, École Polytechnique Fédérale de Lausanne, 1015 Lausanne, Switzerland. *email: maurizio.molinari@irb.usi.ch

The endoplasmic reticulum (ER) is the site of protein, lipid, and oligosaccharide synthesis, calcium storage, and drugs detoxification. Its size (and functions) is maintained at steady state and is adapted to environmental and developmental conditions by a homeostatic equation that comprises the anabolic transcriptional and translational programs of the unfolded protein response (UPR) and the catabolic programs relying on receptor-mediated, lysosomal clearance of select ER subdomains. The UPR increases the size of the ER and its content in resident proteins[1,2]. In contrast, the lysosomal-regulated ER turnover maintains ER size at steady state, prevents excessive ER expansion during ER stress, and, as we recently discovered, regulates ER return at physiologic size during recovery from ER stresses[3,4]. Selective clearance of the ER by endolysosomes (ELs) has originally been observed, in the mid-1960s, during butterfly pupation[5] and remained confined to morphological analyses for over 50 years. It is only with the discovery that lysosomal turnover of the ER is controlled by dedicated ER-resident proteins (FAM134B[6], SEC62[7], RTN3[8], ATL3[9], CCPG1[10], TEX264[11,12] in mammalian cells, Atg39 and Atg40 in yeast[13]) that the studies entered a phase of mechanistic dissection (reviewed in refs. [4,14–18]). All these proteins engage components of the autophagic machinery via their cytosolic domains, which display FIP200-[10] and/or LC3-[6–8,10–12] and/or GABARAP-interacting[9] regions in mammals, and Atg11- and/or Atg8-interacting motifs in yeast[13]. Nutrient deprivation indiscriminately enhances several autophagic pathways to rapidly mobilize amino acids and other cellular building blocks and it has extensively been used to activate and investigate the mechanisms of receptor-mediated ER clearance by mammalian and yeast macro-ER-phagy pathways[6,8–13]. Starvation-induced mammalian macro-ER-phagy relies on engulfment of ER subdomains decorated with FAM134B[6], RTN3[8], ATL3[9], CCPG1[10], and TEX264[11,12] by double-membrane autophagosomes. These eventually fuse with EL for cargo clearance. However, ER-centric signals do exist that trigger clearance of select ER subdomains on activation of individual receptors as reported for CCPG1-mediated control of ER expansion during ER stress[10], for SEC62-controlled reduction of the ER volume to physiologic state after conclusions of acute ER stresses[7], and for ER-to-lysosome-associated degradation (ERLAD) pathways activated to deliver proteasome-resistant misfolded proteins from the ER to EL for destruction[19–21] (and reviewed in refs. [15,17]). Mechanistic dissection of all these pathways is in its infancy and characterization of signal-specific (ER-centric) activation of individual LC3 receptors at the ER membrane awaits further studies and is assessed here in the case of SEC62-regulated recov-ER-phagy. SEC62 is an essential component of the SEC61 protein translocation machinery, where it acts in a functional complex with SEC63 to promote the post-translational entrance of newly synthesized polypeptides in the ER[22,23]. Notably, the function of SEC62 in selective delivery of ER subdomains to EL for clearance is not activated by nutrient deprivation[12], nor at steady state or during ER stress[7]. Our studies revealed that SEC62 controls delivery of excess ER to RAB7/LAMP1-positive EL for clearance during the recovery phase that follows the conclusion of acute ER stresses. In our experiments, acute ER stresses were triggered on transient perturbation of calcium or redox homeostasis to mimic original observation in liver cells showing lysosomal removal of excess ER after cessation of treatments with antiepileptic drugs such as phenobarbital[24,25] (please refer to the detailed description of the protocols for reversible and non-toxic induction of ER stress in ref. [7] and in the Methods section). SEC62-controlled ER turnover during recovery from ER stress, recov-ER-phagy, can also be induced on SEC62 overexpression or on silencing of SEC63, which

participates in SEC62-containing heterodimers[7,26]. Here we report that in contrast to starvation-induced, receptor-mediated ER-clearance[6,8–13], SEC62-driven ER turnover, which is activated in response to an ER-centric signal, that is, the conclusion of an acute ER stress, does not rely on engagement of the macro-autophagy pathway. Rather, resolution of ER stress activates catabolic processes where RAB7/LAMP1-positive EL directly engulf excess ER subdomains via ESCRT-III-mediated piecemeal micro-ER-phagy.

## Results

**ER subdomains delivery within EL on ER stress resolution.** To characterize the mechanisms of mammalian ER remodeling that re-establish physiologic (pre-stress) condition on resolution of ER stresses, we made use of a previously established protocol for acute induction of ER stress on transient exposure of mouse embryonic fibroblasts (MEFs) to cyclopiazonic acid (CPA)[7], a reversible inhibitor of the sarco/ER calcium ATPase[27]. Western blot analyses show that the level of ER stress marker proteins increases in wild-type (WT) MEFs exposed to CPA (Fig. 1a, lane 1 vs. 2, upper and middle panels for BiP and HERP, respectively) and decreases after interruption of the pharmacologic treatment (lane 3)[7]. For ERAD factors like HERP, return at the pre-stress level relies on the activity of cytosolic proteasomes and other ERAD tuning mechanisms[7,28–30]. For conventional ER-resident chaperones and members of the protein disulfide isomerase superfamily, it relies on SEC62-controlled delivery of excess ER within EL for clearance in catabolic processes collectively defined as recov-ER-phagy[7]. Inactivation of lysosomal hydrolases with bafilomycin A1 (BafA1) delays return of these ER-resident chaperones to the pre-stress level[7] and causes their accumulation within EL displaying RAB7 and LAMP1 at the limiting membrane[7] (Fig. 1b, o, Supplementary Fig. 1a). Recov-ER-phagy is faithfully recapitulated on SEC62 overexpression (Supplementary Fig. 1b) and on release of orphan endogenous SEC62 upon silencing of SEC63 expression[7].

**LC3 lipidation in resolution of ER stress.** LC3 lipidation is required for FAM134B-, RTN3-, CCPG1-, ATL3-, and TEX264-dependent macro-ER-phagy[3,6,8–12,14,18]. Not surprisingly, individual ablation of *Atg4B*, *Atg7*, or *Atg16L1*, three components of the LC3 lipidation machinery (Supplementary Fig. 2)[31], inhibits return of ER stress-induced marker proteins to the pre-stress level (BiP in Fig. 1c, e, g, upper panels, lanes 2 vs. 3). Return of HERP at the pre-stress level, which relies on cytosolic proteasomes[7,28], remains unaffected (Fig. 1c, e, g, middle panels). Consistently, ablation of LC3 lipidation abolishes delivery of ER subdomains within EL during recovery from ER stress (Fig. 1d, f, h, o) and when recov-ER-phagy is recapitulated by overexpression of HA-tagged SEC62 (Supplementary Fig. 1d).

**Autophagosome dispensability to resolve ER stress.** To corroborate the notion that SEC62-driven clearance of excess ER on UPR resolution occurs via macro-ER-phagy, we verified the involvement of the autophagosome biogenesis machinery in recov-ER-phagy. Surprisingly, individual ablation of *Ulk1*, *Ulk2*, *Atg13*, and *Atg14*, which are dispensable for LC3 lipidation (Supplementary Fig. 2) but are required for biogenesis of double-membrane autophagosomes[31–34], does not prevent return of BiP to pre-stress level on ER stress resolution (Fig. 1i, k, m, upper panels, lanes 1 vs. 3). Consistently, inactivation of autophagosome biogenesis does not affect delivery of excess ER within EL during recovery from stress (Fig. 1j, l, n, o) and when recov-ER-phagy is recapitulated by SEC62 overexpression (Supplementary Fig. 1e–g).

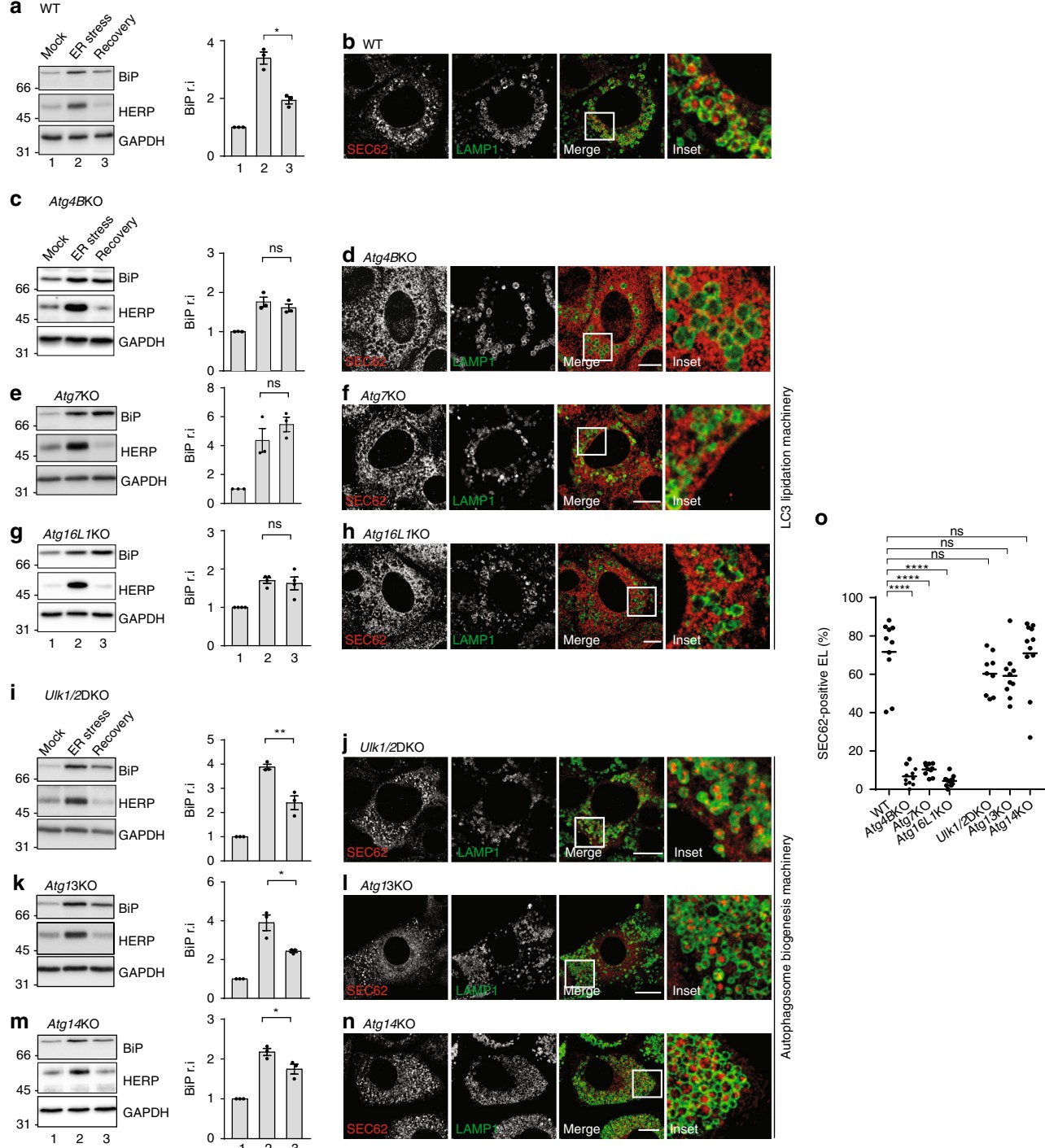

**SNAREs dispensability to resolve ER stress**. To investigate lysosomal delivery of ER subdomains with ultrastructural resolution, we turned to immuno electron microscopy (IEM) and invariably found that ER-derived vesicles (EVs) displaying SEC62 at their limiting membrane (red arrows, Fig. 2a, b) are sequestered by a SEC62-negative membrane (blue arrows, Fig. 2a, b), within the EL (green arrows, Fig. 2a, b). This topology is consistent with macro-ER-phagy, where ER subdomains are captured by double-membrane autophagosomes that eventually fuse with EL to clear their cargo (Fig. 2c), or with micro-ER-phagy, where ER subdomains are directly engulfed by EL (Fig. 2d). A major difference between macro- and micro-autophagy is the requirement for the former an heterotypic membrane fusion event (arrow 2, Fig. 2c,

fusion vs. Fig. 2d, engulfment), which is substantially impaired on ablation of the SNARE proteins STX17 and VAMP8[35,36]. Ablation of STX17 (Fig. 3a, d, f) or of VAMP8 (Fig. 3b, e, f) does not affect delivery of excess ER within EL and does not delay return of chaperones at their pre-stress level during recovery from ER stress (Fig. 3g, h). In agreement with our genetic analyses showing dispensability of autophagosomes and macro-ER-phagy (Figs. 1i–o, 3a–h), IEM analyses do not reveal SEC62-labeled ER fragments within double-membrane autophagosomes when cells are recovering from ER stress. Rather, they show EL caught in the act of capturing SEC62-positive EV by inward invagination of their membranes (Fig. 4a–d and Supplementary Movie 1). Thus, recov-ER-phagy is topologically equivalent to micro-autophagy, a

**Fig. 1** Delivery of endogenous SEC62-labeled EV within EL during recovery from ER stress. **a** Upper panel, WB analysis showing steady-state level of BiP in WT MEF (Mock), BiP induction on cell exposure to CPA (ER stress, lane 2), and return of BiP to the pre-stress level after CPA wash-out (Recovery, lane 3); middle panel, same for HERP; lower panel, GAPDH as a loading control. Quantification of BiP levels in WB, $n = 3$ independent experiments, mean ± SEM, unpaired, two-tailed $t$ test, $*P = 0.0037$. **b** Delivery of SEC62-decorated EV within LAMP1-positive EL in WT MEF during 12 h recovery from an ER stress in the presence of 50 nM BafA1. **c** Same as **a** in Atg4BKO MEF; $n = 3$ independent experiments, mean ± SEM, unpaired, two-tailed $t$ test, $P = 0.4001$. **d** Same as **b** in Atg4BKO MEF. **e** Same as **a** in Atg7KO MEF, $n = 3$ independent experiments, mean ± SEM; unpaired, two-tailed $t$ test, $P = 0.3159$. **f** Same as **b** in Atg7KO MEF. **g** Same as **a** in Atg16L1KO MEF; $n = 4$ independent experiments, mean ± SEM; unpaired, two-tailed $t$ test, $P = 0.6909$. **h** Same as **b** in Atg16L1KO MEF. **i** Same as **a** in Ulk1/2 double-KO MEF; $n = 3$ independent experiments, mean ± SEM; unpaired, two-tailed $t$ test, $**P = 0.0083$. **j** Same as **b** in Ulk1/2 double-KO MEF. **k** Same as **a** in Atg13KO MEF; $n = 3$ independent experiments, mean ± SEM; unpaired, two-tailed $t$ test, $*P = 0.0232$. **l** Same as **b** in Atg13KO MEF. **m** Same as **a** in Atg14KO MEF; $n = 3$ independent experiments, mean ± SEM; unpaired, two-tailed $t$ test, $*P = 0.0497$. **n** Same as **b** in Atg14KO MEF. See Supplementary Fig. 1 for recov-ER-phagy recapitulated on overexpression of SEC62-HA. **o** Quantification of SEC62-positive EV delivery within LAMP1-positive EL ($n = 10, 11, 10, 10, 9, 10, 11$ cells, respectively). One-way analysis of variance (ANOVA) and Dunnett's multiple comparisons test, $^{n.s.}P > 0.05$, $****P < 0.0001$. Molecular weight markers in WB are in kDa. Scale bars for CLSM: 10 µm. WB and IF panels are representative of at least three independent experiments

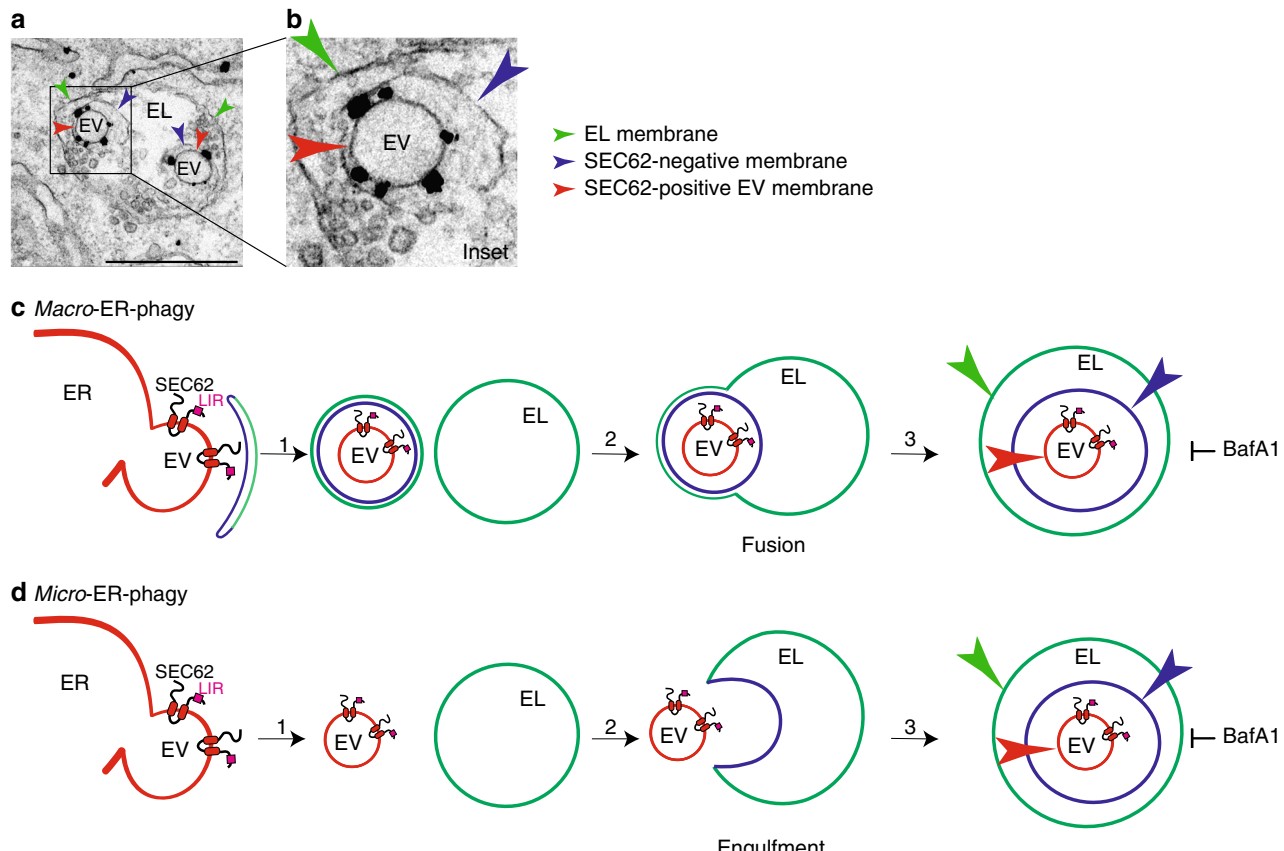

**Fig. 2** Distribution of gold-labeled, endogenous SEC62 by IEM. **a** WT MEFs are exposed 12 h to 10 µM CPA. Recovery is initiated on CPA wash-out and progresses for 12 h in the presence of 50 nM BafA1 to inhibit clearance of EV delivered within EL. **b** Inset of **a**. IEM images are representative of two independent experiments with similar results. **c** Macro-ER-phagy pathway. **d** Micro-ER-phagy pathway. EV, ER-derived vesicle; EL, endolysosome; LIR, LC3-interacting region. Color code of arrows is given in the panels. Scale bar for IEM: 1 µm

poorly characterized type of autophagy involved in clearance of organelles including large ER whorls in yeast[37–40].

**CHMP4B and VPS4A intervention during ER stress resolution.** Inward membrane invagination (i.e., reverse-topology membrane remodeling and scission) is driven by ESCRT-III[41–44]. Consistently, silencing of the charged multivesicular body protein 4B (CHMP4B), an essential ESCRT-III subunit, prevents capture of cytosolic proteins by inward endolysosomal membrane budding[45]. In our experimental setup, silencing of CHMP4B expression (Fig. 5a) substantially inhibits delivery of EV within LAMP1-positive EL during recovery from ER stress

(Fig. 5b–d) and when recov-ER-phagy is recapitulated by SEC62 induction (Fig. 5e–g). Consistently, CHMP4B silencing delays return of ER stress-induced chaperones at their pre-stress level (Fig. 5h, upper panel), without affecting return of HERP (middle panel).

ESCRT-III-driven membrane remodeling and scission relies on the energy delivered on ATP hydrolysis by the auxiliary AAA+ ATPase VPS4A[41–44]. In our experiments, engulfment of SEC62-positive EV by EL during recovery from ER stress was normal in cells expressing fluorescently labeled $VPS4A_{WT}$ (Fig. 6a, merge 1, inset 1, Fig. 6c), but it was substantially inhibited in cells expressing $VPS4A_{K173Q}$, a dominant-negative mutant that cannot bind and hydrolyze ATP[46] (Fig. 6b, merge 1, inset 1, Fig. 6c). The

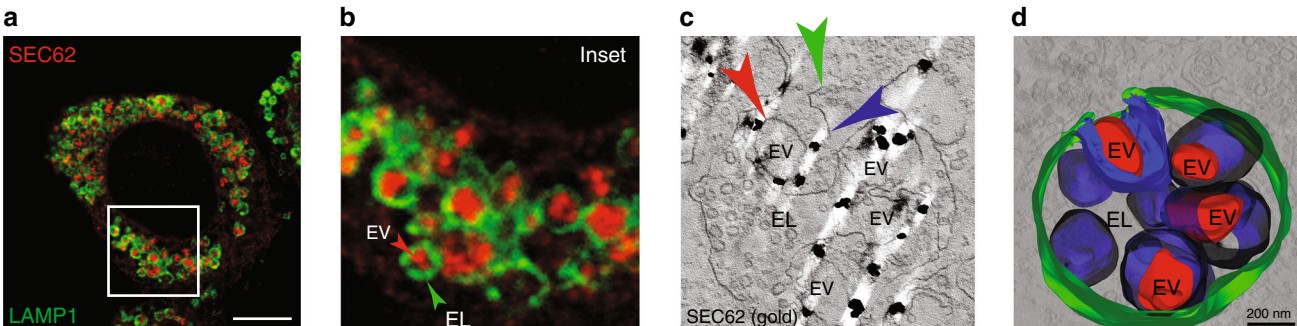

**Fig. 3** SNAREs dispensability for delivery of endogenous SEC62-labeled EV within EL. **a** WB analysis showing the efficiency of STX17 knockout in CRISPR17 MEF (please also refer to ref. [19]). **b** Same as **a** for VAMP8 in CRISPR8 MEF (please also refer to ref. [19]). **c** Delivery of endogenous SEC62-labeled EV within LAMP1-positive EL during recovery from CPA-induced ER stress in WT MEF exposed to 50 nM BafA1 for 12 h. **d** Same as **c** in MEF lacking STX17. **e** Same as **c** in MEF lacking VAMP8. Scale bars: 10 µm. **f** Quantification of EV delivery within EL in **c**–**e** ($n = 15, 12, 10$ cells, respectively). One-way ANOVA and Dunnett's multiple comparisons test, [n.s.]$P > 0.05$. IF panels are representative of three independent experiments. **g** Same as Fig. 1a in cells lacking STX17, $n = 4$ independent experiments, mean ± SEM, unpaired, two-tailed $t$ test, **$P = 0.0083$. **h** Same as Fig. 1a in cells lacking VAMP8; $n = 3$ independent experiments, mean ± SEM, unpaired, two-tailed $t$ test, **$P = 0.0043$. Molecular weight markers in WB are in kDa

**Fig. 4** Engulfment of endogenous SEC62-labeled EV by EL. **a** WT MEF after 12 h recovery from an ER stress in the presence of 50 nM BafA1. IF panel is representative of >10 independent experiments. **b** Inset from **a**. **c** Single slice of an electron tomogram showing the distribution of gold-labeled endogenous SEC62 during recovery from ER stress. IEM image is representative of two independent experiments with similar results. Color code of arrows in **b**, **c** as in Fig. **2a**–**d**. **d** 3D visualization by electron tomography of EL containing EV that display gold-labeled, endogenous SEC62 at the limiting membrane. See also Supplementary Movie 1. Scale bar for CLSM: 10 µm; scale bar for IEM: 200 nm

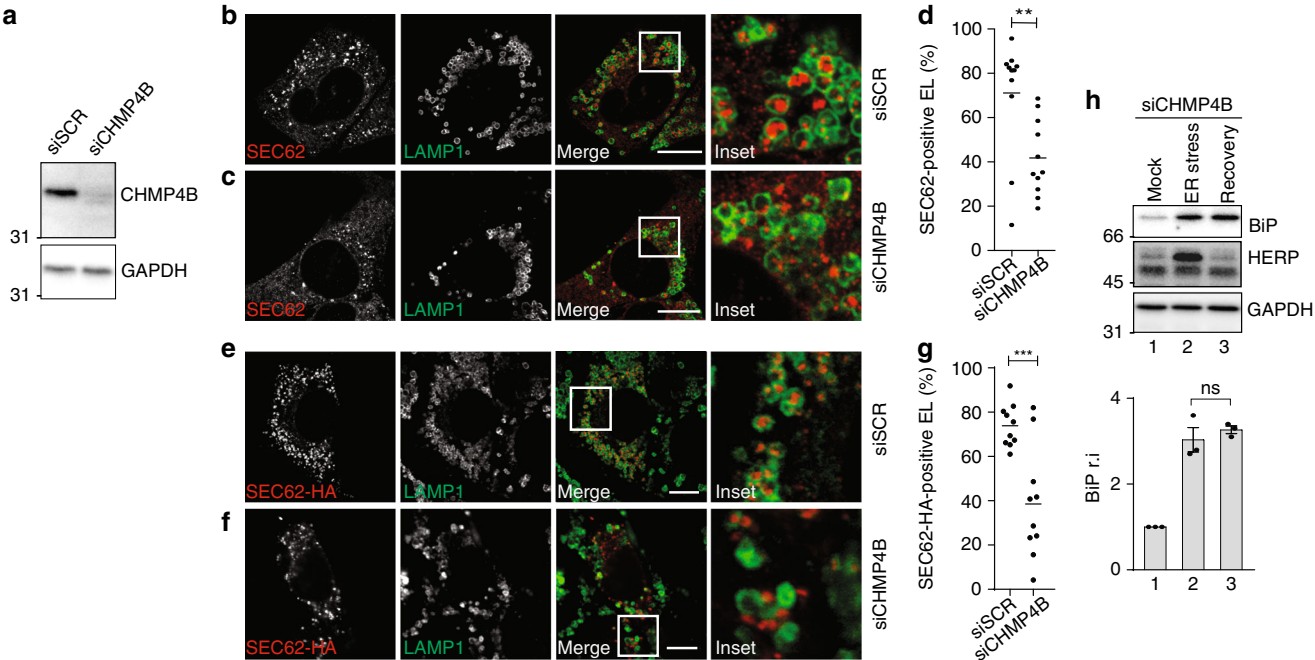

**Fig. 5** A role of CHMP4B in recov-ER-phagy. **a** WB showing knockdown efficiency for siCHMP4B. **b** Same as Fig. 1b in MEF transfected with a scrambled siRNA. **c** Same as **b** in cells transfected with a siRNA silencing CHMP4B expression. **d** Quantification of EV delivery within EL in **b**, **c** ($n = 11$, 11 cells, respectively). Unpaired, two-tailed t test, **$P = 0.0052$. **e** Same as **b** in cells where recov-ER-phagy is recapitulated on SEC62 overexpression. **f** Same as **c** in cells where recov-ER-phagy is recapitulated on SEC62 overexpression. **g** Quantification of EV delivery within EL in **e**, **f** ($n = 10$, 10 cells, respectively). Unpaired t test, ***$P = 0.0006$. Data are representative of at least two independent experiments. Scale bars: 10 μm. **h** Same as Fig. 1a in cells transfected with a siRNA silencing CHMP4B expression. Quantification of BiP levels in WB, $n = 3$ independent experiments, mean ± SEM; unpaired, two-tailed t test, $P = 0.4776$. Molecular weight markers in WB are in kDa; n.s. not significant

role of VPS4A in EV engulfment by LAMP1-positive EL was confirmed on overexpression of SEC62-HA to faithfully recapitulate recov-ER-phagy (Fig. 6d–g). Ultrastructural analyses of cells where the engulfment of excess ER by LAMP1-positive EL is inhibited on inactivation of the ESCRT-III machinery show that SEC62-positive EV remain in close proximity of the EL and are not delivered within the degradative organelles (Fig. 6h–k, Supplementary Movie 2). Notably, VPS4A_WT, which drives EV engulfment, accumulates within LAMP1-positive EL on inactivation of proteolytic enzymes with BafA1 (Fig. 6a, merge 2, inset 2, Fig. 6e, merge 2, white arrows in inset 2 and in inset VPS4A). The inactive VPS4A_K173Q is not found within EL (Fig. 6b, merge 2, inset 2, Fig. 6f, merge 2, red arrows in inset 2 and in inset VPS4A). This is consistent with partitioning of VPS4A within the inward budding structure after the ESCRT-III mediated scission event[44]. Finally, we confirm VPS4A-dependent engulfment of excess ER by LAMP1-positive EL by HaloTag pulse chase (Fig. 7), a protocol for time-resolved analyses of EV segregation recently developed in our lab[19]. Briefly, to monitor by time-resolved fluorescence microscopy the sequential steps of ER subdomains delivery within EL for clearance, WT MEF were transfected with SEC62-HaloTag (Supplementary Fig. 3) and with VPS4A_WT (Fig. 7a, c) or VPS4A_K173Q (Fig. 7b, c). The fate of newly synthesized SEC62-HaloTag is followed by pulsing cells, for 15 min, with the fluorescent HaloTag ligand PBI 5030 (ref. [19] legend of Fig. 7 and Materials and methods). Initially (0–2 h chase), SEC62-HaloTag is not visible within LAMP1-positive EL. Only in cells expressing VPS4A_WT, it is eventually delivered within the EL, where it progressively accumulates on EL inactivation with BafA1 (Fig. 7a, c, 5–12 h chase). In cells expressing inactive VPS4A_K173Q, SEC62-HaloTag remains virtually excluded from the EL throughout the chase (Fig. 7b, c).

**Endogenous LC3B decorates EV**. In macro-autophagy, LC3 is lipidated on the phagophore membrane and eventually recruits cargo within double-membrane autophagosomes[31]. Since autophagosome biogenesis is dispensable for ER turnover at the end of acute ER stresses, we assessed LC3 localization during recov-ER-phagy. Our analyses reveal that endogenous LC3 co-localizes with SEC62-labeled ER accumulating within LAMP1-positive EL on inhibition of lysosomal activity with BafA1 (Fig. 8a). The inhibition of EV engulfment by LAMP1-positive EL on expression of the inactive variant of VPS4A reveals the presence of endogenous LC3B on SEC62-labeled structures both in WT MEF (Fig. 8b) and in MEF with defective autophagosome biogenesis (Fig. 8c, d). The vesicular nature of the SEC62/LC3-positive ER-derived structures that deliver excess ER within EL during recovery from ER stress is confirmed by orthogonal sections of deconvoluted images (Fig. 8f). The formation of EV is abolished in cells expressing a SEC62LIR variant that cannot bind LC3[7] (Fig. 8g) and in Atg7KO MEF with defective LC3 lipidation (Fig. 8e, h). All in all, these results support the notion that the ER-resident LC3-binding protein SEC62, LC3 lipidation and the VPS4A-powered ESCRT-III machinery, but not autophagosome biogenesis, regulate ER fragmentation and piecemeal micro-ER-phagy characterizing the ER turnover on ER stress resolution.

## Discussion

The ER is a plastic organelle of eukaryotic cells, whose volume and activities are adapted in response to intra- and extracellular signals. The anabolic pathways that enlarge the ER and increase its content in resident proteins and enzymes are collectively defined as UPR and have been characterized in molecular details[1,2]. The catabolic pathways that reduce ER volume and

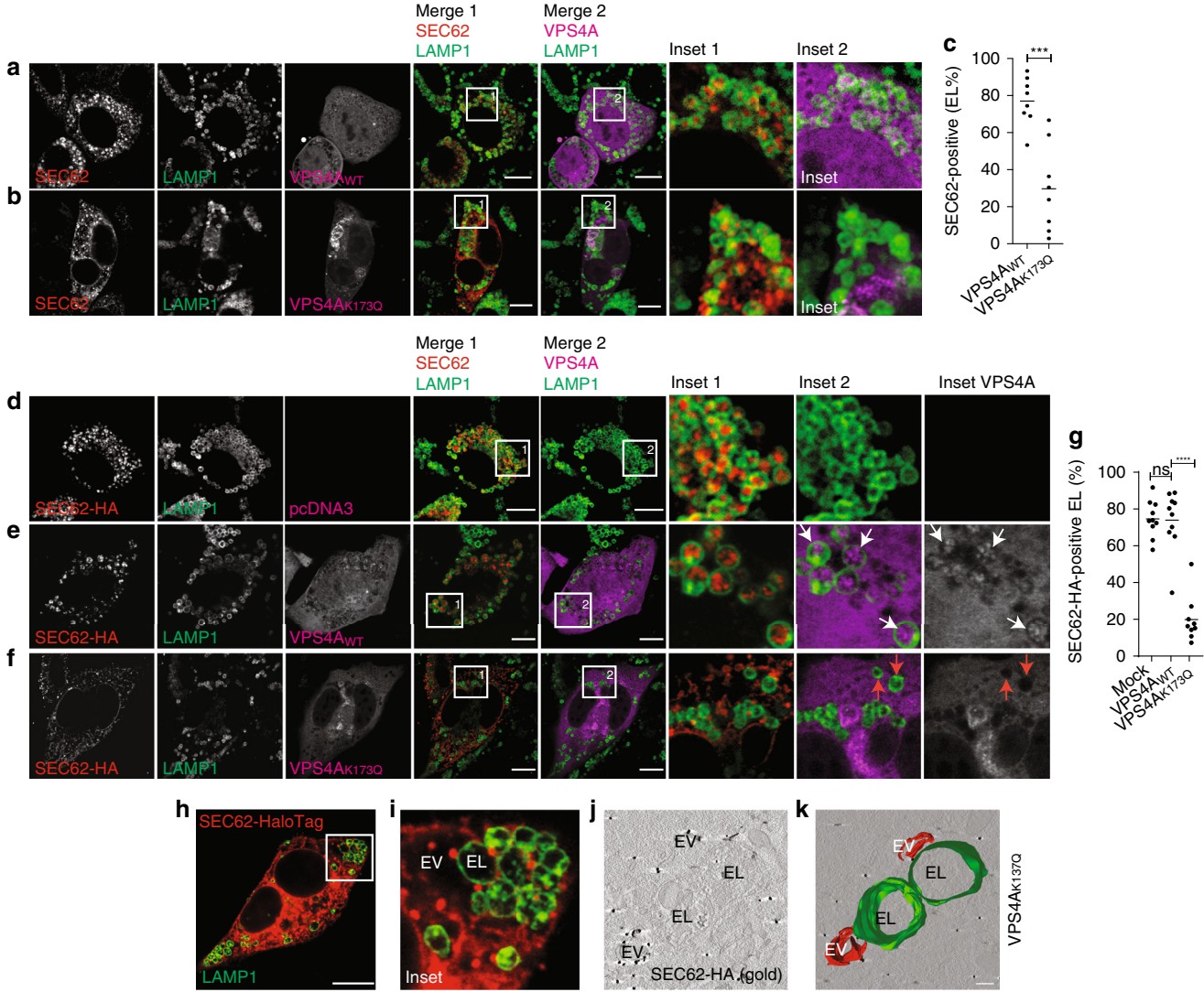

**Fig. 6** A role of VPS4A in recov-ER-phagy. **a** Same as Fig. 1b in MEF transfected with a plasmid for expression of GFP-VPS4A (VPS4A_WT). **b** Same as **a** in cells transfected with a plasmid for expression of GFP-VPS4A_K137Q (VPS4A_K137Q). **c** Quantification of EV delivery within EL in **a**, **b** ($n = 8$, 8 cells, respectively). Unpaired, two-tailed $t$ test, ***$P = 0.0002$. Data are representative of three independent experiments with similar results. **d** Delivery of SEC62-HA within LAMP1-positive EL in MEF transfected with an empty plasmid exposed to BafA1 for 12 h. **e** Same as **d** in cells transfected with a plasmid for expression of GFP-VPS4A_WT. **f** Same as **d** in cells transfected with a plasmid for expression of GFP-VPS4A_K137Q. **g** Quantification of EV delivery within EL in **d–f** ($n = 10$, 10, 9 cells, respectively); ANOVA and Dunnett's multiple comparisons test, n.s.$P > 0.05$, ****$P < 0.0001$. Data are representative of at least three independent experiments. **h** SEC62-HaloTag-positive EV outside LAMP1-positive EL in cells expressing VPS4A_K137Q. See also Supplementary Fig. 3. **i** inset of **h**. Scale bars: 10 μm. **j** Single slice of an electron tomogram. **k** 3D visualization by electron tomography of SEC62-positive EV adjacent to EL. For **j**, **k**, IEM images are representative of two independent experiments with similar results. Scale bar: 250 nm. See Supplementary Movie 2

content by delivering ER subdomains to acidic compartment for destruction are much less understood. The identification of several ER-resident proteins that may engage, upon appropriate stimuli, cytosolic factors such as LC3s/Atg8, GABARAPs, and FIP200/Atg11 to promote delivery of ER fragments within EL/vacuole for clearance has opened the run to dissect receptor-mediated, lysosomal-controlled, ER turnover pathways. At the end of an ER stress, the reduction of the ER volume and of the ER content in resident proteins to pre-stress, physiologic status is crucial to re-establish ER homeostasis. In this phase, cells must, among other things, also re-gain the capacity to produce the secretOME (i.e., the 40% of the proteOME destined to the organelles of the secretory and endocytic compartments, the plasma membrane, and the extracellular space), whose production is temporarily halted during the ER stress phase[47]. The secretOME enters the ER co-translationally via the SEC61 protein

translocation machinery that is endowed with a SEC62:SEC63 functional complex to facilitate access of those polypeptides that are synthesized in the cytosol and imported only after completion of the polypeptide chain via post-translational translocation[22,23]. We recently reported that conclusion of ER stresses activates the LC3-binding function of SEC62 and triggers lysosomal turnover of ER subdomains containing folding factors, but lacking ERAD factors via catabolic pathways collectively defined as recov-ER-phagy[7]. The finding that selective ER delivery to EL for clearance is recapitulated by SEC62 overexpression and by SEC63 silencing[7] hint at possible mutually exclusive role of SEC62 in ER protein import vs. receptor-mediated, selective ER turnover. Notably, the LC3-binding region is dispensable for the former and required for the latter function of SEC62 (ref. [7] and Supplementary Fig. 3c). It will be of interest to verify whether the changes in newly synthesized proteins getting access to the ER during steady state

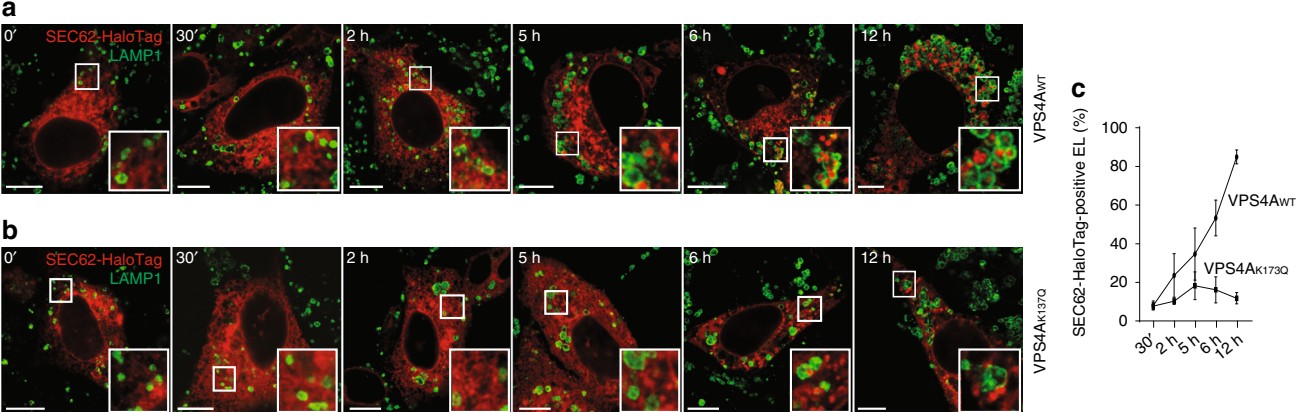

**Fig. 7** Time-resolved analyses of VPS4A-dependent delivery of EV within EL. **a** Cells expressing SEC62-HaloTag and VPS4A$_{WT}$ are incubated for 30 min with 6-chlorohexanol, a non-fluorescent HaloTag ligand that quenches fluorescent detection of the SEC62-HaloTag chimera. Cells are subsequently incubated with a cell-permeable fluorescent HaloTag ligand (PBI 5030) to label, for 15 min, newly synthesized SEC62-HaloTag (fluorescent pulse). Cells are fixed at different times after interruption of the fluorescent pulse on PBI 5030 replacement with 6-chlorohexanol (0 to 12 h[19]) to monitor the progressive delivery of SEC62-HaloTag-labeled EV within LAMP1-positive EL. **b** Same as **a** in MEF expressing SEC62-HaloTag and VPS4A$_{K137Q}$, where EV delivery within EL is virtually abolished. **c** Quantification of SEC62-HaloTag positive EL (VPS4A$_{WT}$ $n = 3, 3, 3, 4, 4$; VPS4A$_{K173Q}$ $n = 3, 3, 4, 4, 4$ cells for the corresponding time points). Mean ± SEM. IF panels are representative of two independent experiments. Scale bar: 10 μm

(presumably more cargo) vs. ER stress (presumably more ER-resident proteins) vs. recovery from ER stress (back to more cargo proteins import?) phases results in variations in the SEC61:SEC62:SEC63 oligomeric composition[22,23] and if this in turn generates orphan SEC62 to promote ER turnover. Alternatively, the triggering signal for a switch in SEC62 function from protein translocation to ER turnover could be a change in the luminal fraction of free BiP (which is regulated in complex manner[48–50]). In fact, SEC63 is a BiP-binding protein and BiP association could set SEC62 free for its role in recov-ER-phagy. All these issues will be assessed in future work.

Here, we report that lipidation of LC3 is required, but the autophagosome biogenesis machinery and macro-ER-phagy are dispensable for clearance of excess ER characterizing the recovery phase from acute ER stresses. As such, recov-ER-phagy, which is triggered by an ER-centric signal, is mechanistically different from starvation-induced turnover of the mammalian ER, which activates (unselective?) macro-autophagic clearance of ER subdomains decorated with FAM134B[6], RTN3[8], ATL3[9], CCPG1[10], TEX264[11,12]. Rather, SEC62-driven return of ER size and content at pre-stress, physiologic status, entails the LC3 lipidation machinery, the ESCRT-III component CHMP4B and the accessory AAA$^+$ ATPase VPS4A. During recovery from ER stress, CHMP4B and VPS4A, whose involvement in physiologic and pathogen-induced membrane repair, remodeling and fission events has been reported[43,44,51], ensure the inward budding of the EL membrane required for the engulfment and clearance of pre-formed ER vesicles that display SEC62 and LC3 at the limiting membrane and contain excess ER chaperones and membranes generated during the phase of ER stress. The function of LC3 in recov-ER-phagy (and in other types of receptor-mediated ER turnover by lysosomes/vacuole) as well as the mode of recruitment of the ESCRT-III machinery that welcomes the incoming ER-derived vesicle containing material to be cleared from cells remains a matter for further studies. However, these data highlight the variety of pathways that cells can activate in response to pleiotropic and to ER-centric signals to ensure lysosomal turnover of the ER. For the mechanistic dissection of catabolic regulation of ER function via lysosomal-controlled ER turnover, it seems crucial to examine in detail cellular responses to ER-centric signals that may activate individual ER-resident LC3-binding proteins. ER-centric signals activate client-specific autophagic and not

autophagic pathways that deliver fragmented ER subdomains containing defective material (certainly misfolded proteins[15,17], possibly aberrant lipids or ER-resident proteins, whose activity is determined post-translationally by regulated turnover[29]) to the lysosomal/vacuolar compartments. Significant on this line, is the characterization of ERLAD pathways that ensure disposal of ER or of ER-exit site subdomains containing misfolded proteins that cannot be dislocated across the membrane for proteasomal degradation[15,17]. Emerging evidence reveal the involvement in ERLAD of mammalian ER-resident LC3-binding proteins such as FAM134B[19,20] and CCPG1[10] and the yeast ER-resident Atg8-binding protein Atg40[52]. These receptors operate conventionally to engage autophagosomes in macro-ER-phagy-like processes[20,52], or unconventionally by ensuring direct delivery of the ER sub-domain to be cleared from cells to EL via vesicular trafficking[19] or via poorly characterized micro-autophagy-like processes[21].

Back to recov-ER-phagy, the broader, possible implications of our findings relate to the fact that amplification of the SEC62 gene and the consequent enhanced ER turnover confers a stress tolerance that correlates with resistance to anti-cancer therapies to a number of carcinomas, including breast, prostate, thyroid, lung adenocarcinoma, and head and neck squamous cell carcinoma[53–55]. Our characterization of the catabolic pathways activated by cells recovering from ER stress to remodel the ER paves the way for identification of therapeutic targets to treat diseases caused by impaired proteostatic control and to unravel in more detail the novel function of the ESCRT-III/VPS4A machinery in membrane remodeling to better define molecular aspects of organelle and membrane dynamics.

## Materials and methods

**Antibodies, expression plasmids, and chemicals**. Antibodies against CNX (Western blot (WB) 1:3000, immunofluorescence (IF) 1:100), CHMP4B (1:1000), SEC62 (WB 1:1000, IF 1:100), and ERj3 (1:250) are kind gifts from A. Helenius, H. Stenmark, and R. Zimmermann. Plasmid encoding GFP-RAB7 is a kind gift from T. Johansen. Commercial antibodies used in this study are from Stressgen (BiP, 1:1000), Chondrex (HERP, 1:1500), Developmental Studies Hybridoma Bank (LAMP1, 1:50), Sigma and Santa Cruz (HA, WB 1:3000, IF 1:100), Merck Millipore (GAPDH, 1:30,000), Sigma (STX17, 1:1000), Abcam (VAMP8, 1:1000), Santa Cruz (Actin 1:500), Novus and Sigma (LC3B, WB 1:1000, IF 1:50), Promega (HaloTag, 1:1000), and MBL (p62, 1:1500). The secondary horse radish peroxidase (HRP)-conjugated antibodies were from Jackson Immunoresearch (rabbit, 1:10,000), Santa Cruz (goat, 1:20,000), and SouthernBiotech (mouse, 1:20,000). Alexa Fluor-conjugated secondary antibodies (1:300) from Invitrogen, Jackson

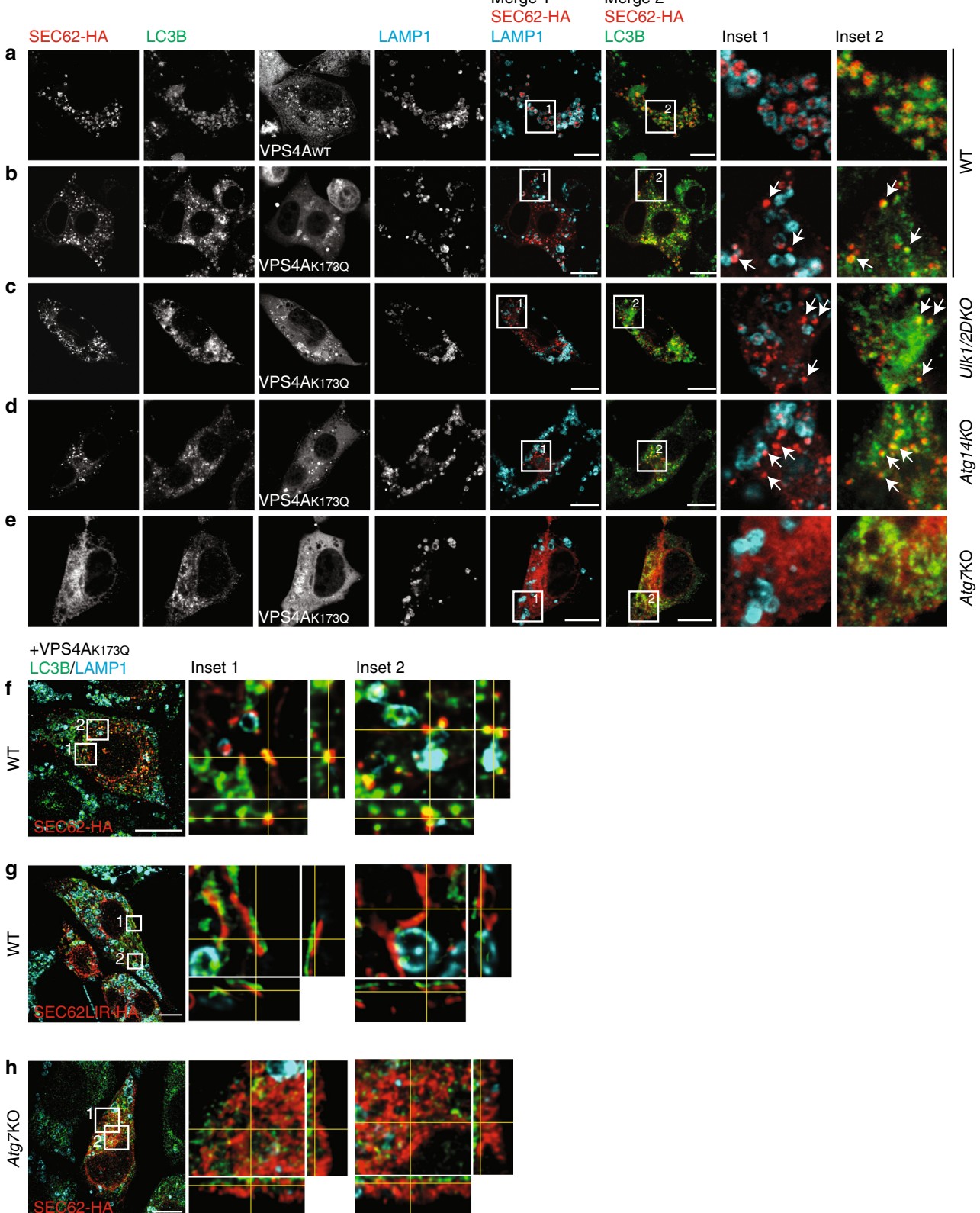

**Fig. 8** LC3B localization on SEC62-positive EV. **a** SEC62-HA co-localizes with endogenous LC3B inside LAMP1-positive EL in WT MEF-expressing VPS4A$_{WT}$ exposed to BafA1 for 12 h. **b** SEC62-HA/LC3B-positive EV (white arrows) fails to be delivered inside the EL in WT MEF-expressing VPS4A$_{K173Q}$. **c** Same as **b** in *Ulk1/2* double-KO MEF. **d** Same as **b** in *Atg14*KO MEF. **e** Same as **b** in *Atg7*KO MEF. **f** Max projection of a WT cell expressing SEC62-HA and VPS4A$_{K173Q}$ after deconvolution. Insets show orthogonal sections of selected region. **g** Same as in **f** with SEC62LIR-HA. **h** Same as **f** in *Atg7*KO MEF cells. Scale bars: 10 μm. IF panels are representative of at least two independent experiments

Immunoresearch, Thermo Fisher, and BioLegend. Plasmids encoding GFP-VPS4A$_{WT}$ and GFP-VPS4A$_{K173Q}$ were a kind gift from John McCullough. SEC62 was subcloned in a pcDNA3.1 expression plasmid with the addition of a C-terminal HA-tag or HaloTag[7]. CPA (Sigma) and BafA1 (Sigma) were used for 12 h, if not otherwise specified, at final concentrations of 10 μM and 50 nM, respectively. PS341 (Millennium Pharmaceuticals) was used for 8 h at a final concentration of 10 μM.

**Induction of transient ER stress**. To induce a mild and reversible ER stress, cultured cells were exposed for up to 12 h to CPA, a reversible inhibitor of the sarco/ER calcium ATPase[7,27]. An alternative protocol implies cell exposure to dithiothreitol[7] (DTT, a reversible perturbator of redox homeostasis[56]), whereas other ER stress-inducing drugs such as tunicamycin and thapsigargin, which are irreversible inhibitors of GlcNAc phosphotransferase[57] and of the sarco/ER calcium ATPase[58], respectively, are toxic[7]. Consistent with UPR induction, CPA treatment caused splicing of XBP1 transcripts, 25% attenuation of global protein synthesis, and induction of ER stress marker transcripts and proteins[7]. The catabolic events characterizing recovery from ER stress (i.e., the delivery of ER subdomains within EL for clearance) were investigated after CPA wash-out as specified in the figure legends and in ref.[7].

**Cell culture, transient transfection, and RNA interference**. MEF and HEK cells were grown at 37 °C and 5% CO$_2$ in Dulbecco's modified Eagle's medium (DMEM) medium supplemented with 10% fetal calf serum (FCS). WT and Atg7KO MEF cells were gifts from M. Komatsu. Atg4BKO, Atg13KO, Atg14 and Atg16L1KO, and Ulk1/2DKO MEF cells were kindly provided by G. Marino, F. Reggiori, T. Saitoh, and S. Tooze, respectively, and have been characterized in ref.[19] and in Supplementary Fig. 2. To induce autophagy, cells were washed three times with Earle's balanced salt solution (EBSS, Thermo Fisher) and then incubated in EBSS for 2 h in the presence or absence of 100 nM BafA1.

Transient transfections were performed using JetPrime transfection reagent (PolyPlus) according to the manufacturer's instructions. RNA interferences were performed in MEF cells plated at 50–60% confluence. Cells were transfected with scrambled small interfering RNA or small interfering RNA (siRNA) against CHMP4B (5′-AAACAGUCCCUCUACCAAAtt-3′, 50 nM per dish, Silencer Select Pre-designed, Ambion). Cells were processed for immunofluorescence or for biochemical analyses 48 h after transfection (see below).

**Preparation of KO cells**. STX17- and VAMP8-KO MEF and SEC62-KO HEK cells were generated by CRISPR/Cas9 genome editing. For the generation of the guideRNA-Cas9 plasmids, lentiCRISPRv2-puro system (Addgene52961) was obtained from Addgene (http://www.addgene.org). Guide sequences were obtained from the Cas9 target design tools (crispr.mit.edu:8079 and/or www.addgene.org/pooled-library). All protocols and information can be found at the website https://www.addgene.org/crispr. The target sequences for guide RNA were synthesized by Microsynth. Two annealed oligonucleotides (5′-CACCGCTGTGGTTGACTACTGCAAC-3′, 5′-AAACGTTGCAGTAGTCAACCACAGC-3′ for human SEC62; 5′-GCGCTCCAATATCCGAGAAA-3′, 5′-TTTCTCGGATATTGGGAGCGC-3′ for murine STX17; 5′-CCACCTCCGAAACAAGACAG-3′, 5′-CTGTCTTGTTTCGGAGGTGG-3′ for murine VAMP8) were inserted into the lentiCRISPRv2-puro vector using the *Bsm*BI restriction site. Vectors were transfected in HEK and MEF cells with JetPrime (Polyplus) according to the manufacturer's instructions[7,19]. Cells were cultured in DMEM supplemented with 10% FBS. Two days after transfection, the medium was supplemented with 2 μg/ml puromycin. After 10 days, puromycin-resistant clones were picked and gene KO was verified by WB (ref.[7] for SEC62; Fig. 3a, b and ref.[19] for STX17 and VAMP8) and with a translocation assay for SEC62 (ref.[7] and Supplementary Fig. 3c).

**Cell lysis, Western blot**. Cells were washed in ice cold phosphate-buffered saline (PBS) containing 20 mM N-ethylmaleimide (NEM) and lysed in 1% sodium dodecyl sulfate (SDS) in HEPES-buffered saline, pH 6.8, or with RIPA buffer (1% Triton X-100, 0.1% SDS, 0.5% sodium deoxycholate in HBS, pH 7.4) containing 20 mM NEM and protease inhibitor cocktail (1 mM PMSF, chymostatin, leupeptin, antipain, and pepstatin, 10 μg/ml each). Post-nuclear supernatants were collected by centrifugation at 10,000 × g for 10 min at 4 °C. Samples were denatured and reduced in DTT-containing sample buffer for 5 min at 95 °C and separated by SDS-PAGE (polyacrylamide gel electrophoresis). Proteins were transferred to polyvinylidene fluoride membranes with the Trans-Blot Turbo Transfer System (Bio-Rad). Membranes were blocked 10 min with 10% (w/v) non-fat dry milk (Bio-Rad) and stained with the above-mentioned primary antibodies for 90 min and for 45 min with HRP-conjugated secondary antibodies. Membranes were developed using the Luminata Forte ECL detection system (Millipore) and signals were acquired with the ImageQuant LAS 4000 system (GE Healthcare Life Sciences) or with the Amersham Imager 680 system. Image quantifications were performed with the Multi Gauge Analysis tool (Fujifilm). Membrane stripping was performed using Re-Blot Plus Strong Solution (Millipore) following the manufacturer's instructions. Uncropped blots can be found as Supplementary Fig. 4.

**Confocal laser scanning microscopy**. MEF cells plated on alcian blue-coated glass coverslips were treated according to the experimental setup. Cells were washed twice in PBS and fixed at room temperature for 20 min with 3.7% formaldehyde in PBS or 5 min in 100% methanol at −20 °C for endogenous LC3B detection. Cells were then incubated for 15 min with permeabilization solution (PS, 0.05% saponin, 10% goat serum, 10 mM HEPES, 15 mM glycine, pH 7.4) to improve antigen accessibility. Cells were incubated with the primary antibodies diluted 1:50–1:200 in PS for 90 min, washed for 15 min in PS, and then incubated with Alexa Fluor-conjugated secondary antibodies diluted 1:300 in PS for 30 min. Cells were rinsed with PS and water and mounted with Vectashield (Vector Laboratories) supplemented with 4′,6-diamidino-2-phenylindole. Confocal pictures were acquired using a Leica TCS SP5 microscope with a 63.0 × 1.40 Oil UV objective. FIJI was used for image analysis and processing. Figure 8f–h were acquired with LEICA HCX PL APO CS 100.0 × 1.44 Oil UV objective with a XY pixel size of 50 nm and Z step of 125 nm and pinhole 0.8 AU. Images were deconvoluted with Autoquant 3.1.1 (Media Cybernetics) with a spherical aberration correction.

**Immunogold electron microscopy**. Cells were plated on alcian blue-coated glass coverslips and fixed 10 min with 0.05% glutaraldehyde in 4% paraformaldehyde (PFA) EM grade and 0.2 M HEPES buffer and 50 min in 4% PFA EM grade in 0.2 M HEPES buffer. After three washes in PBS, cells were incubated 10 min with 50 mM glycine and blocked 1 h in blocking buffer (0.2% bovine serum albumin, 5% goat serum, 50 mM NH$_4$Cl, 0.1% saponin, 20 mM PO$_4$ buffer, 150 mM NaCl). Staining with primary antibodies and nanogold-labeled secondary antibodies (Nanoprobes) were performed in blocking buffer at room temperature. Cells were fixed 30 min in 1% glutaraldehyde and nanogold was enlarged with gold enhancement solution (Nanoprobes) according to the manufacturer's instructions. Cells were post fixed with osmium tetroxide, embedded in epon, and processed into ultrathin slices. After contrasting with uranyl acetate and lead citrate, the sections were analyzed with Zeiss LEO 512 electron microscope. Images were acquired by 2k × 2k bottom-mounted slow-scan Proscan camera controlled by the EsivisionPro 3.2 software.

**HaloTag pulse-chase analyses**. MEF cells were plated on alcian blue-coated glass coverslip and transfected with SEC62-HaloTag and VPS4$_{WT}$ or VPS4$_{K173Q}$. Seventeen hours after transfection, cells were incubated with 15 μM 6-chlorohexanol (Sigma) in DMEM 10% FCS for 30 min. 6-Chlorohexanol is cell-permeable black ligand that irreversibly binds the SEC62-HaloTag-binding pocket. After three washes in DMEM 10% FCS, cells are pulsed 15 min with 1 μM of the fluorescent ligand PBI 5030 (Promega), which exclusively enters the HaloTag ligand binding pocket of newly synthesized SEC62-HaloTag, and 100 nM BafA1 in DMEM 10% FCS. The fluorescent ligand is removed and after three washes in DMEM 10% FCS, cells are incubated with 15 μM of the black ligand to block incorporation of the fluorescent ligand in the newly synthetized SEC62-HaloTag, and 100 nM BafA1 in DMEM 10% FCS. Cells were fixed at increasing time points (0, 30 min, 2, 5, 6, and 12 h) and processed for confocal laser scanning microscopy as described above. HaloTag pulse chase has been described in ref.[19].

**Statistical analyses and reproducibility**. In panels showing WB or IF, unless stated otherwise, images are representative of three independent experiments with similar results. The number of independent experiments or cell numbers is given in the figure's legend. Statistical analyses were performed only if sample size was ≥3 using GraphPad Prism 7 software. One-way analysis of variance with Dunnett's multiple comparisons test or unpaired, two-tailed $t$ test were used to asses statistical significance. $P$ values are given in the figure legends; $^{n.s}P < 0.05$; $^*P < 0.05$; $^{**}P < 0.01$; $^{***}P < 0.001$; $^{****}P < 0.0001$.

**Reporting summary**. Further information on research design is available in the Nature Research Reporting Summary linked to this article.

## Data availability
The data that support the findings of this study are available from the authors on reasonable request. The source data underlying the figures can be found in the Source Data file.

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

## Acknowledgements

We thank G. Brambilla Pisoni, F. Fumagalli, A. Helenius, M. Komatsu, G. Marino, N. Mizushima, J. Noack, P. Paganetti, F. Reggiori, T. Saitoh, C. Settembre, S. Tooze, X. Wang, and the members of Molinari's lab for gift of reagents, discussions, and critical reading of the manuscript. We are grateful to the ALEMBIC facility at San Raffaele Scientific Institute, Milan, Italy for the help in electron microscopy analyses. M.M. is supported by Signora Alessandra, AlphaONE Foundation, Foundation for Research on Neurodegenerative Diseases, Swiss National Science Foundation, and Comel and Gelu Foundations.

## Author contributions

Conceptualization: M.M. together with M.L.; methodology: M.L., A.R., D.M., and M.M.; investigation: M.L., A.R., D.M., and M.M.; writing original draft: M.M.; supervision: M.M.

## Competing interests

The authors declare no competing interests.
