## [Peer Review File · Nature Communications]

Reviewers' comments:

Reviewer #1 (Remarks to the Author):

ER-phagy is an exciting and important area of cell biology. A number of ER-phagy mechanisms have been identified, including one previously described by the Molinari lab. In this previous study, they showed that SEC62 was involved in returning the ER to basal state after stress via lysosomal degradation of ER portions filled with specific luminal content (redundant chaperone proteins etc.). This "recovER-phagy" process was presumed to involve macroautophagy (at least by the field at large), given the involvement of SEC62 as an apparent LC3-binding "cargo receptor". In fact, this was only the second potential ER-phagy pathway characterised in mammals, and constituted a landmark paper. Here, Loi et al significantly expand on these findings by showing that SEC62 does not participate, after all, in conventional autophagy (macroautophagy) of the ER. SEC62 instead stimulates piecemeal micro-ER-phagy, the direct lysosomal engulfment of preformed ER vesicles. This is an unexpected finding and constitutes a new pathway for ER-phagy and ER homeostasis in mammals. Indeed, this finding is of high importance to the fields of ER biology and autophagy, not least as it says that current models for ER-phagy function are incomplete.

The manuscript is well written and cogent. The data are elegantly and simply presented, outlining succinctly the key observation. Insofar as they go, they are technically sound.

My sole recommendation would be that the authors provide a small amount of further mechanistic insight, notwithstanding any unforeseen technical issues or misinterpretations associated with the following suggestions (which should be rebutted). In particular, it would seem incumbent on the authors to address the role of LC3-binding by SEC62. Are the cytosolic ER-derived vesicles (EVs) – which can now be "trapped" prior to lysosomal engulfment (for example by expressing mutant versions of VPS4A, as seen in Fig. 4h) - labelled with LC3? Do the EVs fail to form, or merely fail to recruit LC3, in the absence of ATG7? This latter question would help address where the SEC62-LC3 interaction might act. To complement this, the authors could also interrogate exactly where it is that the recovER-phagy process stalls when LIR mutant (non-LC3 binding) SEC62 rather than wild-type SEC62 is used – do EVs form in the first place, do they label with LC3, do they become engulfed or not?

MINOR POINTS

- As written, I could not ascertain how quantification of SEC62 localisation within LAMP2 vesicles was performed. Are the $n \sim 10$ cells counted per independent experiment (and then one representative independent experiment shown) or counted over the \geq three independent experiments described. For example, see description of quantification within legend to Figure 1.

- The statement in the abstract that recovER-phagy is the only pathway for selective mammalian ER clearance that does not involve autophagosomes may be slightly confusing. The mechanism of mutant alpha-1-antitrypsin turnover from (presumably) ER-derived vesicles by lysosomal fusion (rather than engulfment), described by some of the authors of this paper in Fregno et al (EMBO J 2018), may constitute one such example (albeit no entry of ER membrane into the lysosomal lumen).

- Description of the UPR and of UPR resolution may benefit from an extra line or so in the Introduction.

Reviewer #2 (Remarks to the Author):

Loi et al. follow up on the same group's 2016 characterization of Sec62-dependent "recov-ER-phagy" in MEF cells. The experimental procedures, which are largely imaging, follow those that were established the previous 2016 paper from the same group in NCB. Overall, the quality of the imaging is quite good. The paper takes the next natural steps in defining much, if not all, of the machinery of recov-ER-phagy, which includes the LC3 coat of autophagy and the LC3 conjugation machinery, but not upstream components of canonical autophagy, and also includes the ESCRTs that are involved in microautophagy and other topologically related membrane scission processes. There are a number of directions in which the paper could be extended. Some obvious questions are what recruits and triggers LC3 conjugation in the absence of the ULK1 complex, what recruits CHMP4 and VPS4, and what signals the onset of recovery from ER stress. The function of LC3 is not clear and it would be interesting to know which LIR proteins are involved, and whether any of the ESCRTs have functional LIR motifs. Nevertheless, the findings are already quite a significant addition to the field, and they seem to be fully justified by the data as presented.

My only comment for revision is that the ms. is so concise and compressed that it is very hard to follow, and too much of the data are in the supplementary information. The authors should rewrite it in as a normal article instead of the ultra-compressed letter format. The extended data can mostly be moved back to the main manuscript.

Reviewer #3 (Remarks to the Author):

In this manuscript, the authors use a genetic approach to define the specific proteins involved in restoring ER balance in response to ER stress. Previous work showed that following an acute ER stress, cells promote ER-phagy through a mechanism involving the LC3 binding activity of the ER protein Sec62. However, the specific downstream mechanism of this process required further definition. Here, the authors show that ablation of core autophagy genes involved in macro-ER phagy do not block ER stress resolution downstream of Sec6, suggesting that this process does not proceed through macro-ER-phagy. This is further supported by genetic evidence showing that deletion of SNARE proteins involved in membrane fusion (specifically required for macro-ER-phagy) do not block ER resolution observed following acute ER stress. Instead, these results are most consistent with micro-ER-phagy involving engulfment of ER-derived vesicles as opposed to fusion. Supporting this model, deletion of CHMP4 – an ESCRT-III protein important for lysosomal engulfment of vesicles – blocks macro-ER-phagy following acute ER stress. Similarly, overexpression of catalytically inactive variants of the AAA+ VPS4 – a protein centrally involved in ESCRT-mediated membrane remodeling – also blocks micro-ER-phagy observed following acute ER stress. Overall, these results show that recovery of ER content following acute ER insults proceeds through micro-ER-phagy through a process involving the ESCRT-III components CHMP4B and VPS4A.

The included data does seem fairly convincing for demonstrating the importance of ESCRT signaling in mammalian ER resolution induced following acute ER stress. The novelty of this result is somewhat reduced, as previous results showed that the ESCRT pathway (including VPS4) is involved ER micro-autophagy in yeast following an acute lipid stress (a type of stress that induce ER stress)(see Vevea et al (2015) Dev Cell). While the previous work does focus on removal of damaged proteins (not the resolution of ER protein levels to pre-stress levels described herein), in my opinion, the previous links between micro-ER-phagy and ESCRT pathways does take novelty away from the current findings. Apart from the novelty, there are still questions pertaining to the regulation of this process that I think are quite important. For example, what are the signals responsible for directing ER-derived vesicles to micro-ER-phagy following an acute ER stress? Is this pathway regulated by UPR signaling or another mechanism? The authors claim that this is distinct from the basal turnover of the ER afforded by macro-ER-phagy, so what are the signals that direct the micro-ER-phagy events? Despite the nice mechanistic work described herein, considering previous links between micro-ER-phagy and ESCRT observed in yeast, I think that some insights into the regulation of this process is required for this

manuscript to reach a level suitable for publication in a high impact journal such as Nat Comm.

Apart from the above, other significant issues that should be addressed are as below:

1. The authors need to do a better job of putting their work into context with what has been learned about micro-ER-phagy in other systems (notably yeast). As highlighted above, there is evidence for a role of micro-ER-phagy linking to ESCRT in yeast following acute ER stress, which was not sufficiently discussed or referenced.
2. As mentioned above, the authors need to discuss the regulation of this process following acute ER stress. Is this linked to the UPR or another ER stress-regulated pathway? Does the UPR transcriptionally regulate ESCRT-III in mammals? The entire mechanism of regulation does not need to be established in this paper, but some experiments need to be included to discuss regulation since the authors claim that this is distinct from basal ER clearance.
3. Along the same lines as the above, a common experiment required for pathways activated by ER stress is to use other ER stressors to observe similar effects. Does the same type of regulation happen with DTT or tunicamycin – two other types of reversible ER stressors – or is this something that is selective for calcium dysregulators?
4. Apart from showing regulation mechanisms, the authors could also demonstrate some direct disease relevance to these findings in mammalian cells as suggested in the discussion (e.g., showing that can you influence drug resistance in cancer by inhibiting this pathway). I think that this type of demonstration would also increase the overall interest of these findings and highlight the uniqueness of these findings as compared to what has been observed in yeast.

Reviewers' comments:

Reviewer #1 (Remarks to the Author):

ER-phagy is an exciting and important area of cell biology. A number of ER-phagy mechanisms have been identified, including one previously described by the Molinari lab. In this previous study, they showed that SEC62 was involved in returning the ER to basal state after stress via lysosomal degradation of ER portions filled with specific luminal content (redundant chaperone proteins etc.). This “recovER-phagy” process was presumed to involve macroautophagy (at least by the field at large), given the involvement of SEC62 as an apparent LC3-binding “cargo receptor”. In fact, this was only the second potential ER-phagy pathway characterised in mammals, and constituted a landmark paper. Here, Loi et al significantly expand on these findings by showing that SEC62 does not participate, after all, in conventional autophagy (macroautophagy) of the ER. SEC62 instead stimulates piecemeal micro-ER-phagy, the direct lysosomal engulfment of preformed ER vesicles. This is an unexpected finding and constitutes a new pathway for ER-phagy and ER homeostasis in mammals. Indeed, this finding is of high importance to the fields of ER biology and autophagy, not least as it says that current models for ER-phagy function are incomplete.

The manuscript is well written and cogent. The data are elegantly and simply presented, outlining succinctly the key observation. Insofar as they go, they are technically sound.

My sole recommendation would be that the authors provide a small amount of further mechanistic insight, notwithstanding any unforeseen technical issues or misinterpretations associated with the following suggestions (which should be rebutted). In particular, it would seem incumbent on the authors to address the role of LC3-binding by SEC62.

We thank the referee for the insightful comments.

- Are the cytosolic ER-derived vesicles (EVs) – which can now be “trapped” prior to lysosomal engulfment (for example by expressing mutant versions of VPS4A, as seen in Fig. 4h) - labelled with LC3?

The labeling of SEC62-decorated ER-derived vesicles with LC3 is shown in WT, ULK1/2-, ATG14-KO, expressing active or inactive forms of VPS4A in the new Fig. 8A-D, F.

Do the EVs fail to form, or merely fail to recruit LC3, in the absence of ATG7? This latter question would help address where the SEC62-LC3 interaction might act. To complement this, the authors could also interrogate exactly where it is that the recovER-phagy process stalls when LIR mutant (non-LC3 binding) SEC62 rather than wild-type SEC62 is used – do EVs form in the first place, do they label with LC3, do they become engulfed or not?

There is no co-localization between SEC62 and the LC3 structures (Runwal et al 2019) formed in ATG7-KO cells (new Fig. 8E, H). There is no co-localization between SEC62LIR and LC3 in WT cells (new Fig. 8G). At this stage, we have no indication that ER-derived vesicles do form in cells expressing the LIR variant of SEC62 and in cells with defective LC3 lipidation. Certainly, under these conditions the delivery of ER subdomains within EL is substantially impaired (Fig. 1, new Extended Figure 3 and Fumagalli et al 2016).

MINOR POINTS

1- As written, I could not ascertain how quantification of SEC62 localisation within LAMP2 vesicles was performed. Are the $n \sim 10$ cells counted per independent experiment (and then one representative independent experiment shown) or counted over the \geq three independent experiments described. For example, see description of quantification within legend to Figure 1.

This has now been better explained in the new Figure legends.

2- The statement in the abstract that recovER-phagy is the only pathway for selective mammalian ER clearance that does not involve autophagosomes may be slightly confusing. The mechanism of mutant alpha-1-antitrypsin turnover from (presumably) ER-derived vesicles by lysosomal fusion (rather than engulfment), described by some of the authors of this paper in Fregno et al (EMBO J 2018), may constitute one such example (albeit no entry of ER membrane into the lysosomal lumen).

The abstract has been shortened and this sentence has been deleted.

3- Description of the UPR and of UPR resolution may benefit from an extra line or so in the Introduction.

Please also refer to referee 3, question 7. In the introduction, we added “In our experiments, acute ER stresses were triggered on transient perturbation of calcium or redox homeostasis to mimic original observation in liver cells showing lysosomal removal of excess ER after cessation of treatments with antiepileptic drugs such as phenobarbital^{24,25} (please refer to the detailed description of the protocols for reversible and non-toxic induction of ER stress in⁷ and in the Methods section).”

A sub-section in Methods entitled “Induction of transient ER stress” has also been added.

Reviewer #2 (Remarks to the Author):

Loi et al. follow up on the same group's 2016 characterization of Sec62-dependent "recov-ER-phagy" in MEF cells. The experimental procedures, which are largely imaging, follow those that were established the previous 2016 paper from the same group in NCB. Overall, the quality of the imaging is quite good. The paper takes the next natural steps in defining much, if not all, of the machinery of recov-ER-phagy, which includes the LC3 coat of autophagy and the LC3 conjugation machinery, but not upstream components of canonical autophagy, and also includes the ESCRTs that are involved in microautophagy and other topologically related membrane scission processes. There are a number of directions in which the paper could be extended. Some obvious questions are what recruits and triggers LC3 conjugation in the absence of the ULK1 complex, what recruits CHMP4 and VPS4, and what signals the onset of recovery from ER stress. The function of LC3 is not clear and it would be interesting to know which LIR proteins are involved, and whether any of the ESCRTs have functional LIR motifs. Nevertheless, the findings are already quite a significant addition to the field, and they seem to be fully justified by the data as presented.

My only comment for revision is that the ms. is so concise and compressed that it is very hard to follow, and too much of the data are in the supplementary information. The authors should rewrite it in as a normal article instead of the ultra-compressed letter format. The extended data can mostly be moved back to the main manuscript.

We thank the referee for the positive evaluation of our work. As suggested, we expanded the test/explanations and we moved most of the supplementary material in the article.

Reviewer #3 (Remarks to the Author):

In this manuscript, the authors use a genetic approach to define the specific proteins involved in restoring ER balance in response to ER stress. Previous work showed that following an acute ER stress, cells promote ER-phagy through a mechanism involving the LC3 binding activity of the ER protein Sec62. However, the specific downstream mechanism of this process required further definition. Here, the authors show that ablation of core autophagy genes involved in macro-ER phagy do not block ER stress resolution downstream of Sec62, suggesting that this process does not proceed through macro-ER-phagy. This is further supported by genetic evidence showing that deletion of SNARE proteins involved in membrane fusion (specifically required for macro-ER-phagy) do not block ER resolution observed following acute ER stress. Instead, these results are most consistent with micro-ER-phagy involving engulfment of ER-derived vesicles as opposed to fusion. Supporting this model, deletion of CHMP4B – an ESCRT-III protein important for lysosomal engulfment of vesicles – blocks macro-ER-phagy following acute ER stress. Similarly, overexpression of catalytically inactive variants of the AAA+ VPS4 – a protein centrally involved in ESCRT-mediated membrane remodeling – also blocks micro-ER-phagy observed following acute ER stress. Overall, these results show that recovery of ER content following acute ER insults proceeds through micro-ER-phagy through a process involving the ESCRT-III components CHMP4B and VPS4A.

The included data does seem fairly convincing for demonstrating the importance of ESCRT signaling in mammalian ER resolution induced following acute ER stress. The novelty of this result is somewhat reduced, as previous results showed that the ESCRT pathway (including VPS4) is involved ER micro-autophagy in yeast following an acute lipid stress (a type of stress that induce ER stress)(see Vevea et al (2015) Dev Cell). While the previous work does focus on removal of damaged proteins (not the resolution of ER protein levels to pre-stress levels described herein), in my opinion, the previous links between micro-ER-phagy and ESCRT pathways does take novelty away from the current findings.

We respectfully disagree with this comment. The Vevea et al paper does not offer a link between micro-ER-phagy and the ESCRT pathways. The work by Vevea et al (which is not cited in our manuscript) describes biogenesis of lipid droplets and their clearance via *micro*-lipophagy. Micro-lipophagy is an LC3-independent, ESCRT-III mediated engulfment of lipid droplets by the yeast vacuole. Misfolded proteins generated in the ER may remain trapped in the lipid droplets, however, Vavea et al show in Fig. 5 and SFig5 and explicitly write in their paper (p590 and 592) that “... lipid stress induces degradation of lipid droplets markers but not of markers of mitochondria or ER”. All in all, the organelle examined in the Vevea et al paper is different (lipid droplet in their case vs. ER in our work). The pathway described in the Vevea et al paper is also

different as there is no involvement of LC3-binding proteins, of LC3 and of the LC3 lipidation machinery (all of which are involved in the process that we report in our manuscript).

ER turnover via Micro-ER-phagy has been reported in yeast by Schuck et al 2014 (cited in our manuscript), but the mechanism remains to be established (Lipatova et al 2018, cited in our manuscript).

Apart from the novelty, there are still questions pertaining to the regulation of this process that I think are quite important. For example,

1- what are the signals responsible for directing ER-derived vesicles to micro-ER-phagy following an acute ER stress?

This is an important issue and we thank the referee for the question. Recov-ER-phagy can be recapitulated by SEC62 overexpression and by silencing of SEC63 (the functional partner of SEC62 in ER protein translocation). This leads us to propose that generation of free SEC62 (which binds LC3, Fumagalli et al 2016) could be the triggering signal for this type of ER turnover. This is now discussed in the manuscript (page 9).

2- Is this pathway regulated by UPR signaling or another mechanism?

This pathway is not induced by the UPR (Fumagalli et al. 2016, and please refer to answer to questions 6 and 7, below).

3-The authors claim that this is distinct from the basal turnover of the ER afforded by macro-ER-phagy, so what are the signals that direct the micro-ER-phagy events?

See response to question 1.

4- Despite the nice mechanistic work described herein, considering previous links between micro-ER-phagy and ESCRT observed in yeast, I think that some insights into the regulation of this process is required for this manuscript to reach a level suitable for publication in a high impact journal such as Nat Comm.

See comment on the Vevea et al paper written above.

Apart from the above, other significant issues that should be addressed are as below:

5. The authors need to do a better job of putting their work into context with what has been learned about micro-ER-phagy in other systems (notably yeast). As highlighted above, there is evidence for a role of micro-ER-phagy linking to ESCRT in yeast following acute ER stress, which was not sufficiently discussed or referenced.

See comment on the Vevea et al paper written above.

6. As mentioned above, the authors need to discuss the regulation of this process following acute ER stress. Is this linked to the UPR or another ER stress-regulated pathway? Does the UPR transcriptionally regulate ESCRT-III in mammals? The entire mechanism of regulation

does not need to be established in this paper, but some experiments need to be included to discuss regulation since the authors claim that this is distinct from basal ER clearance.

Recov-ER-phagy is not induced by the UPR (Fumagalli et al 2016). However, to answer to the reviewer's question, we determined transcriptome and proteome changes on UPR induction in Bergmann et al JBC 2018. Our data show that SEC62, components of the ESCRT-III machinery and VPS4A are not induced by UPR triggered on cell exposure to Thapsigargin, Tunicamycin, DTT and CPA. Data are available in Bergmann et al JBC 2018 and have been deposited to the GEO genomics data repository with the accession number GSE108346. The mass spectrometry proteomics data have been deposited to the ProteomeXchange Consortium via the PRIDE partner repository with the dataset identifier PXD008529.

7. Along the same lines as the above, a common experiment required for pathways activated by ER stress is to use other ER stressors to observe similar effects. Does the same type of regulation happen with DTT or tunicamycin – two other types of reversible ER stressors – or is this something that is selective for calcium dysregulators?

To answer this question and question 3 by referee 1, we have modified the text as reported below. Importantly, recov-ER-phagy is not activated by ER stress. The fact that the pathway is activated during recovery from ER stress, makes it crucial that cells are exposed to ER stresses which are reversible and non-toxic. The identification of protocols to induce reversible and non-toxic ER stresses by exposing mammalian cells to CPA or DTT is a crucial part of the Fumagalli et al paper 2016. The manuscript now contains a sentence in the Intro “In our experiments, acute ER stresses were triggered on transient perturbation of calcium or redox homeostasis to mimic original observation in liver cells showing lysosomal removal of excess ER after cessation of treatments with antiepileptic drugs such as phenobarbital ^{24,25} (please refer to the detailed description of the protocols for reversible and non-toxic induction of ER stress in ⁷ and in the Methods section).”

A sub-section in Methods has also been added. “Induction of transient ER stress-To induce a mild and reversible ER stress, cultured cell were exposed for up to 12 h to cyclopiazonic acid (CPA), a *reversible* inhibitor of the sarco/ER calcium ATPase ^{7,26}. An alternative protocol implies cell exposure to dithiothreitol ⁷ (DTT, a *reversible* perturbator of redox homeostasis ⁵⁶), whereas other ER stress-inducing drugs such as tunicamycin and thapsigargin, which are *irreversible* inhibitors of GlcNAc phosphotransferase ⁵⁷ and of the sarco/ER calcium ATPase ⁵⁸, respectively, are toxic ⁷. Consistent with UPR induction, CPA treatment caused splicing of XBP1 transcripts, 25% attenuation of global protein synthesis, induction of ER stress marker transcripts and proteins ⁷. The catabolic events characterizing recovery from ER stress (i.e., the delivery of ER subdomains within EL for clearance) were investigated after CPA wash-out as specified in the figure legends and in ⁷.”

8. Apart from showing regulation mechanisms, the authors could also demonstrate some direct disease relevance to these findings in mammalian cells as suggested in the discussion (e.g., showing that can you influence drug resistance in cancer by inhibiting this pathway). I think that this type of demonstration would also increase the overall interest of these findings and highlight the uniqueness of these findings as compared to what has been observed in yeast.

See comment on the Vevea et al paper written above. We agree with the referee that showing that inhibition of recov-ER-phagy influence cancer progression would be a great achievement. However, this is beyond the scope of this manuscript.

REVIEWERS' COMMENTS:

Reviewer #1 (Remarks to the Author):

Loi et al have provided additional data on the function of the SEC62-LC3 interaction in the new mammalian microER-phagy pathway. They have thus addressed my only query of the initial submission. The manuscript remains extremely timely and the autophagy field would benefit from rapid publication of this work.